# Modulatory Effect of Cucurbitacin D from *Elaeocarpus hainanensis* on ZNF217 Oncogene Expression in NPM-Mutated Acute Myeloid Leukemia

**DOI:** 10.3390/ph17121561

**Published:** 2024-11-21

**Authors:** Sabrina Adorisio, Alessandra Fierabracci, Ba Thi Cham, Vu Dinh Hoang, Nguyen Thi Thuy Linh, Le Thi Hong Nhung, Maria Paola Martelli, Emira Ayroldi, Simona Ronchetti, Lucrezia Rosati, Silvia Di Giacomo, Trinh Thi Thuy, Domenico Vittorio Delfino

**Affiliations:** 1Section of Pharmacology, Department of Medicine and Surgery, University of Perugia, 06129 Perugia, Italy; adorisiosabrina@libero.it (S.A.); emira.ayroldi@unipg.it (E.A.); simona.ronchetti@unipg.it (S.R.); lucrezia.rosati@dottorandi.unipg.it (L.R.); 2Bambino Gesù Children’s Hospital, IRCCS, 00165 Rome, Italy; alessandra.fierabracci@opbg.net; 3Department of Chemistry, Graduate University of Science and Technology, Vietnam Academy of Science and Technology (VAST), Hanoi 10072, Vietnam; chamhoasinh@gmail.com (B.T.C.); thuylinh1992.bk@gmail.com (N.T.T.L.); thuy@ich.vast.vn (T.T.T.); 4Institute of Chemistry, VAST, Hanoi 10072, Vietnam; 5School of Chemistry and Life Sciences, Hanoi University of Science and Technology, Hanoi 10000, Vietnam; hoang.vudinh@hust.edu.vn; 6Faculty of Chemical Technology, Hanoi University of Industry, Hanoi 10000, Vietnam; nhunglth82@gmail.com; 7Hematology, Department of Medicine and Surgery, University of Perugia and ‘Santa Maria della Misericordia’ Perugia Hospital, 06123 Perugia, Italy; maria.martelli@unipg.it; 8Department of Physiology and Pharmacology “V. Erspamer”, Sapienza University of Rome, P.le Aldo Moro 5, 00185 Rome, Italy; silvia.digiacomo@uniroma1.it; 9Department of Food Safety, Nutrition and Veterinary Public Health, Italian National Institute of Health, 00161 Rome, Italy; 10Foligno Nursing School and Master in Physiotherapy in Musculoskeletal and Rheumatological Area, Department of Medicine and Surgery, University of Perugia, 06129 Perugia, Italy

**Keywords:** *Elaeocarpus hainanensis*, cucurbitacin D, ZNF217, nucleophosmin, acute myeloid leukemia

## Abstract

**Background/Objectives:** The expression of oncogene zinc-finger protein 217 (ZNF217) has been reported to play a central role in cancer development, resistance, and recurrence. Therefore, targeting ZNF217 has been proposed as a possible strategy to fight cancer, and there has been much research on compounds that can target ZNF217. The present work investigates the chemo-preventive properties of cucurbitacin D, a compound with a broad range of anticancer effects, in hematological cancer cells, specifically with regard to its ability to modulate ZNF217 expression. **Methods:** Different cucurbitacins were isolated from the Vietnamese plant *Elaeocarpus hainanensis*. The purified compounds were tested on nucleophosmin-mutated acute myeloid leukemia and other hematological cancer cell lines to assess their effects on the cell cycle, cell viability and apoptosis, and the expression of ZNF217. **Results:** Cucurbitacin D resulted in a reduction in the number of acute myeloid leukemia cells by inducing an increase in apoptosis and blocking cell cycle progression. It also led to a significant decrease in ZNF217 expression in the nucleophosmin-mutated acute myeloid leukemia cell line but not in the other hematologic cancer cell lines. The reduction in ZNF217 expression contributed significantly to the blocking of cell cycle progression but did not affect apoptosis. **Conclusions:** The obtained results suggest that cucurbitacin D is a promising molecule for targeting mutated nucleophosmin or its pathway in acute myeloid leukemia cells, although further studies are needed for in-depth investigations into its specific mechanisms.

## 1. Introduction

Multiple factors contribute to cancer development, resistance, and recurrence [1]. In particular, the accumulation of genetic aberrations in crucial genes is a prominent contributing factor that leads to uncontrolled intracellular signaling networks [2]. Among the genes identified so far [1,3], the oncogene zinc-finger protein 217 (*ZNF217*) has recently emerged as a potential therapeutic target. ZNF217 protein has been reported to foster tumorigenesis through several mechanisms, including sustaining cell proliferation, enabling replicative immortality, and resisting cell death [4]. In the later stages of cancer, it also induces epithelial–mesenchymal transition and chemotherapy resistance [5]. The deleterious effects of ZNF217 are believed to involve the dysregulation of several pathways, including the phosphatidylinositol 3-kinase/protein kinase B (PI3K/AKT), mitogen-activated protein kinase (MAPK), Janus kinase/signal transducers and activators of transcription 3 (JAK/STAT3), and transforming growth factor beta (TGF-β) pathways [5]. Aberrant expression of ZNF217 has been highlighted in several tumors, such as breast, colon, liver, and thyroid tumors, and was found to be correlated to therapeutic resistance and poor patient outcomes [5,6]. Therefore, targeting ZNF217 could be a promising strategy to fight cancer and its related chemoresistance. So far, only one drug, triciribine, has been shown to counteract ZNF217-driven deleterious effects [7]. Discovering new compounds that can target ZNF217 would help improve the prognosis of patients with cancer.

Plants represent an important source of novel bioactive compounds that can help in the prevention and treatment of cancer and act as chemo-preventive agents [8]. The consumption of plant-based bioactive compounds can prevent cancer occurrence in healthy individuals, avoid the progression of premalignant lesions in high-risk patients, and have benefits as adjuvant treatment in oncologic or post-treatment patients [9]. Cucurbitacins are a group of recently emerged plant-based compounds that have shown potential chemo-preventive effects. Cucurbitacins are highly oxidized tetracyclic triterpenoids distributed across several plant families (e.g., Cucurbitaceae and Elaeocarpaceae) that act as defense compounds [10]. They have attracted attention due to their broad range of pharmacological effects, which include anticancer and chemosensitizing effects [11]. So far, several cucurbitacins have been isolated [12], among which Cucurbitacin B, E, and I represent the most frequently investigated compounds [13]. In addition, cucurbitacin D (CucD) also deserves attention on account of its antiproliferative activity in a variety of cancers, including human adult T-cell leukemia and breast, cervical, prostate, gastric, lung, and liver cancers, by modulating several pathways involved in cancer cell proliferation, such as the PI3K/AKT/mTOR, MAPKs, JAK2/STAT3, ROS/p38, NF-kappa B, and EGFR pathways, which are linked to ZNF217 expression [14,15,16,17,18]. Despite this, the modulatory effect of cucurbitacin D on ZNF217 expression has not been investigated yet.

Based on the gaps in the literature described above, the objectives of the present study are to evaluate whether the chemo-preventive properties of cucurbitacins are related to the modulation of ZNF217 expression in acute myeloid leukemia (AML) and other hematological cancer cell lines that have been shown to express ZNF217. To this end, CucD, 3-epi-isocucurbitacin D (IsocucD), and cucurbitacin I (CucI) were isolated from the aerial parts of the Vietnamese plant *Elaeocarpus hainanensis* Oliv., an ornamental plant commonly used in Asian urban landscapes and gardens [19,20]. Despite the scarcity of information about the use of *E. hainanensis* in ethnopharmacology, it has been in use in oriental medicine [19]. For example, the use of *E. hainanensis* in traditional medical practices has been reported in Institutum Botanicum Academiae Sinicae, Iconographia Cormophytorum Sinicorum, and Flora of China. Indeed, the recovery of cucurbitacins from the pruning process could represent an interesting recycling strategy via which they could be obtained for therapeutic purposes without endangering the ecosystem. This approach is also important considering the difficulties involved in the synthesis of CucD owing to its complex chemical structure [21]. In this study, the purified cucurbitacins were used to treat nucleophosmin (NPM)-mutated AML and other hematological cancer cell lines in order to examine their effects on viability, apoptosis, and cell cycle progression. In addition, the influence of cucurbitacins on the expression of ZNF217 was evaluated. The significance of the present study lies in the identification of physiologically active compounds from terrestrial plants and in the pharmacological characterization of compounds of natural origin.

## 2. Results

### 2.1. Isolation of Cucurbitacins from E. hainanensis

Compounds **1**, **2**, and **3** (namely, CucD, IsocucD, and a mixture of CucD and CucI, respectively) were isolated from the methanolic extract of dried twigs and leaves of E. hainanensis by using combined column chromatographic separations with appropriate mobile phases. Based on NMR spectroscopic analysis of the structures of the compounds and comparison with previously reported reference values, compounds **1**, **2**, and **3** were identified as CucD [22,23], IsocucD [23], and a mixture of CucD + CucI [22,23,24], respectively (see detailed NMR data in Appendix A in the Appendix A). The ratio of CucD and CucI in compound **3** was 1:1 and was determined by proton integral intensity (H-6).

### 2.2. Effect of CucD on the Number and Survival of AML Cells

Cells of the AML cell line OCI-AML3 (OCI) were cultured with different concentrations of CucD, IsocucD, or the CucD + CucI mixture for 24 h at 37 °C and then collected and counted. Figure 1 shows the chemical structures (on the left side) and the number of harvested cells (right side). All three cucurbitacins resulted in a significant reduction in the number of OCI cells, but CucD had the most significant effect at a minimum effective concentration of 0.3 μg/mL. Since the effects were already evident at 24 h, we did not carry out any time-course experiments.

### 2.3. Effect of CucD and IsocucD on Apoptosis of OCI-AML3 Cells

The reduction in cell number induced by the isolated compounds may be the result of increased cell death, decreased cell proliferation, or both. Therefore, we first investigated the induction of apoptotic cell death by the two most powerful compounds, namely CucD and IsocucD. We stained the cell nuclei with propidium iodide (PI) and performed cytofluorimeter analysis [25]. The results showed that the same concentrations that resulted in a reduction in cell number (0.58 and 5.805 μM for CucD; 1.899 and 18.99 μM for IsocucD) also promoted a significant increase in apoptosis (Figure 2A,B). Thus, the reduction in the number of OCI-AML3 cells induced by both cucurbitacins was attributable, at least in part, to an increase in apoptosis.

The increase in apoptosis induced by CucD and IsocucD prompted us to investigate the underlying molecular mechanisms. We cultured OCI-AML3 cells for 24 h with the control compound, CucD, or IsocucD and then extracted RNA for real-time analyses. Significant differences were calculated based on the results of five independent experiments. Figure 2 shows significant downregulation of the pro-apoptotic molecule TNF-α and significant downregulation of the anti-apoptotic molecule Bcl2. These results indicate that the TNF-α-dependent extrinsic apoptotic pathway was inhibited by CucD and IsocucD treatment, whereas the Bcl2-dependent intrinsic apoptotic pathway was induced by treatment with CucD and IsocucD.

### 2.4. Effect of CucD and IsocucD on the Cell Cycle of OCI-AML3 Cells

Another possible cause of the observed reduction in cell number is that cucurbitacins can interfere with cell cycle progression and, consequently, with cell proliferation. To investigate this, OCI cells were treated with the tested compounds for 24 h and then collected, stained with PI, and analyzed by flow cytometry. Figure 3A (quantitative analysis) and Figure 3B (representative experiment) show that, at concentrations of 0.58 and 5.805 μM, CucD caused a significant reduction in the percentage of cells in the G0/G1 phase (left bar panel) and in the S phase (middle bar panel), with a significant increase in the percentage of cells in the G2/M phase (right panel). A slight difference was seen with IsocucD, since this compound caused a significant decrease in the number of cells in the G0/G1 phase when used at a concentration of 1.899 μM but only caused a small, although significant, increase at 18.99 μM (C, left panel). IsocucD, similar to CucD, caused a significant decrease in the number of cells in the S phase at both effective concentrations (C, middle panel), whereas it caused an increase in the number of cells in the G2/M phase at 1.899 µM but not at 18.99 μM (C, right panel). Therefore, the tested compounds caused not only an increase in apoptosis but also blockage of their entry into the cell cycle (phase G0/G1) and blockage of DNA synthesis (S phase), with consequent accumulation of cells in the G2/M phase (mitosis). These results suggest that the tested compounds decreased cell number by acting on both the survival and proliferation of leukemia cells. Based on the results of the previous experiments, we chose the minimum effective concentration for all subsequent experiments.

### 2.5. Effects of CucD and IsoCucD on OCI-AML2 Cells

CucD and IsoCucD were also tested on an additional acute myeloid leukemia cell line, OCI-AML2, which differs from OCI-AML3 because it is not mutated on an NPM gene. Both substances significantly decreased the number of cells (Figure 4A). CucD and IsoCucD also increased the percentage of apoptotic cells, although the increase was not significant (Figure 4B). When tested on cell cycle progression in OCI-AML2 cells, CucD significantly decreased the number of cells in G0/G1 and increased the number of cells in G2/M phases of the cell cycle, whereas the effect of IsoCucD was not significant (Figure 4C). Thus, CucD affected the number of OCI-AML2 acting only on the cell cycle progression.

### 2.6. Effects of CucD and IsocucD on Cell Proliferation Pathways

We next analyzed the potential mechanisms by which CucD and IsocucD affected the cell cycle by using Western blotting analysis to measure the expression of p21 in CucD- or IsocucD-treated OCI cells. Protein bands from Western blots of five independent experiments were quantitated, and as shown in Figure 5, both CucD and IsocucD were found to significantly upregulate p21 at the tested concentrations. Because p21 is regulated by p53 [18,19], we also investigated whether CucD or IsocucD treatment induces p53 expression in the OCI cells. We found that neither compound affected the expression of p53. This means that the p21-dependent pathway is, at least, partially involved in the effect of CucD and IsocucD on cell cycle arrest. We also investigated the potential roles of ERK and p38 in CucD- and IsocucD-induced p21 expression using Western blotting. Figure 5 shows that the levels of phosphorylated ERK were significantly decreased after CucD treatment. While IsocucD treatment also resulted in a decrease in ERK phosphorylation, it was not significant compared to the effect of DMSO treatment (control). The Western blots show two bands for phosphorylated ERK that represent its two isoforms (41/43 KDa). Conversely, the levels of phosphorylated p38 were significantly increased after both CucD and IsocucD treatment. Thus, the CucD- and IsocucD-dependent changes in p21 expression in OCI cells are possibly related, at least in part, to changes in MAPK pathway signaling.

### 2.7. Effect of CucD and IsocucD on the Expression of the ZNF217 Gene

We tested the effects of cucurbitacins on ZNF217 after screening for ZNF217 and other genes that are currently under study in our lab. In the subsequent experiments, we investigated its possible modulation of the OCI-AML3 cell line by culturing the cells with CucD or IsocucD for 24 h in four independent experiments and then collecting them for RNA and protein extraction. ZNF217 expression was assessed by both real-time PCR and Western blotting. As can be seen from Figure 6A, ZNF217 mRNA was expressed by OCI-AML3 cells, and both CucD and IsocucD significantly downregulated its expression at their minimum effective concentration (0.3 and 0.5 μg/mL for CucD and IsocucD, respectively). These results were confirmed using Western blotting experiments (Figure 6B). Indeed, the ZNF217 protein was expressed by OCI-AML3 cells, and both compounds were able to significantly decrease its expression, with CucD being more effective than IsocucD.

### 2.8. Effect of CucD and IsocucD on Other Cancer Cell Lines

To understand if the expression of ZNF217 and its modulation by the tested compounds were limited to OCI cells, three independent experiments were performed on four additional cell lines, namely the chronic lymphatic leukemia cell lines PGA1 and MEC1, the monocytic lymphoma cell line U937, and the T-cell lymphoma cell line Jurkat. The cells were cultured in the absence and presence of CucD or IsocucD. After 24 h, they were collected, and the protein component was extracted for Western blotting analysis. As shown in Figure 7, all the cell lines expressed ZNF217, but, contrary to the observations in the OCI cell line, neither of the compounds caused a decrease in ZNF217 expression. Thus, while ZNF217 is expressed by cancer cells of various types, the tested cucurbitacins selectively decreased ZNF217 expression only in the AML cell line OCI-AML3. Notably, in experiments performed to compare the effects of CucD and IsoCucD in NPM-wild type OCI-AML2 cells versus NPM-mutated OCI-AML3, we found that only CucD significantly decreased the expression of ZNF217, and this effect was only seen in the NPM-mutated OCI-AML3 cells but not in the NPM-wild type OCI-AML2 cells, thus indicating that the effect of CucD on ZNF217 decrease was linked to the mutation of NPM gene (Figure 8).

### 2.9. Effect of ZNF217 Silencing on OCI-AML3 Cells Treated with the Isolated Cucurbitacins

In the subsequent experiments, we tried to determine whether the downregulation of ZNF217 by the tested substances was responsible for the pro-apoptotic and anti-proliferative effects observed. The ZNF217 gene was significantly silenced after 24 and 72 h of siRNA treatment (Figure 9A,B) and was associated with a significant, although slight, decrease in cell number after 72 h (Figure 9C), which was not attributable to an increase in apoptosis (Figure 9D) but, rather, to a significant increase in the number of cells in the G0/G1 phase (Figure 9E: left panel, 48 and 72 h of treatment) and a decrease in the number of cells in the S and G2/M phases (middle and left panel, respectively). Thus, these experiments suggest that the downregulation of ZNF217 is responsible, at least in part, only for cell cycle blocking but not for the pro-apoptotic action of CucD and IsocucD.

## 3. Discussion

The anti-tumor activity of cucurbitacin derivatives has been extensively studied in the context of AML. In this work, we report a new target gene of cucurbitacins, namely the oncogene ZNF217, in AML cells. To the best of our knowledge, this gene, which is known to be involved in the development of tumors and their metastatic spread, has not been documented in these cancer cells. Our findings demonstrate that cucurbitacins cause significant downregulation of ZNF217. This observation is pertinent because, so far, only one other molecule with this effect has been documented. Furthermore, this effect was only observed in AML cells with an NPM mutation, which is associated with the development of AML in 30% of the cases.

The ZNF217 protein belongs to the Kruppel-like family of zinc finger transcription factors. It has several functions in cancer cells, such as promotion of proliferation, evasion of growth suppressors, replicative immortality capacity, resistance to apoptosis, enrichment of cancer stem cells, drug resistance, and activation of invasion and metastasis. Its oncogenic functions are believed to be the result of an excessive increase in its expression. Indeed, its aberrant expression has been highlighted in several tumors, such as breast, colon, liver, and thyroid tumors [5]. Moreover, patients with ZNF217-positive tumors experience poor outcomes, with a worse relapse-free survival and overall survival than those with ZNF217-negative tumors [6]. Therefore, targeting the ZNF217 oncogene could be promising not only in terms of treating cancer and its related chemoresistance but also as a potential biomarker for its early diagnosis.

Based on the data described above, in the present study, we have analyzed the effect of cucurbitacins isolated from E. hainanensis on the expression of the ZNF217 oncogene in AML and other leukemia cells. Indeed, the preliminary experiments allowed us to identify for the first time the expression of the ZNF217 oncogene in AML cells. We further went on to evaluate its modulation as a potential mechanism underlying the antiproliferative effects of cucurbitacins on AML cells.

At first, cucurbitacin D, 3-epi-isocucurbitacin D, or a mixture of cucurbitacin D and cucurbitacin I (see also Appendix A) were tested in AML cells and found to result in a significant reduction in the number of OCI-AML3 cells after 24 h of treatment. Among them, CucD exhibited the most potent effect at the lowest concentration of 0.3 μg/mL (change μM). Subsequent experiments were carried out only on CucD and IsocucD, with a focus on their ability to induce increased apoptosis and/or inhibition of proliferation. Both factors exhibited significant effects, with CucD increasing apoptosis by about 33% and IsocucD increasing apoptosis by about 50%. Apoptosis is activated by at least two different pathways. The mitochondrial (i.e., intrinsic) pathway leads to downregulation of anti-apoptotic molecules, such as Bcl2, with the sequential release of cytochrome c from mitochondria and activation of caspase-9, which directly cleaves and activates caspase-3. The second (i.e., extrinsic) pathway involves activation of caspase-8 that is triggered by the stimulation of death receptors, such as TNFR, by its ligand TNF-α [26]. It is unclear if the anti-apoptotic effect of the tested substances is attributable to their effect on the intrinsic or extrinsic pathways, since the downregulation of the anti-apoptotic protein Bcl-2, which belongs to the intrinsic pathway, after 24 h of treatment may be the result of the apoptotic process and not an indicator of the direct anti-apoptotic effect of the tested compounds. Similarly, it is difficult to draw definitive conclusions based on the significant decrease in TNF-α expression. TNF-α, in fact, binds to TNFR, a type I transmembrane protein that belongs to the tumor necrosis factor/nerve growth factor receptor family [27], and the triggering of TNFR activates the extrinsic apoptosis pathway. Further, we have evaluated mRNA levels and not protein expression, so it is difficult to make a conclusion about whether the protein expression of TNF-α was also affected. Future studies assessing caspase-8 activation could help clarify whether the extrinsic apoptosis pathway is activated by the tested substances.

Cell cycle analysis highlighted the ability of both substances to block cell cycle progression as they caused a significant decrease in the number of cells in the G0/G1 and S phases and accumulation of cells in the G2/M phase. Thus, the reduction in the number of OCI-AML3 cells was due to both an increase in cell death and cell cycle blockage. Analysis of cell cycle pathways showed that CucD and IsocucD caused an increase in the expression of p21 but not p53. Further, they caused an increase in the phosphorylation of p38. It has been reported that phosphorylation of members of the MAPK pathway, such as p38, increase p21 stability [26], whereas decrease in ERK phosphorylation decreases p21 stability by promoting its degradation [27]. Therefore, cucurbitacins, mainly CucD, were able to decrease the number of AML cells by targeting both the apoptosis and proliferation ability of these cells.

Previous studies have investigated the potential of CucD and IsocucD in the management of cancer, although the potential of IsocucD has been reported to a lesser extent. In particular, CucD and IsocucD were found to exert antiproliferative activity in MCF7 breast cancer cells by partly disrupting Hsp90 client protein maturation [28]. CucD has been shown to block the cell cycle and to induce antiproliferative effects in endometrial and ovarian cancer cells by modulating pro- and antiapoptotic factors [29]. Similar properties of CucD have also been reported in cervical cancer cells and were associated with the inhibition of STAT3 activation [30]. In pancreatic cancer cells, CucD induced cell cycle arrest and apoptosis by triggering ROS generation and activation of the p38 MAPK pathway [15]. Further, the CucD-mediated downregulation of genes and proteins belonging to the PI3K/AKT/mTOR, MAPK, and JAK2/STAT3 cascades has been highlighted in liver cancer [14]. Along with the proapoptotic effects of CucD, it has been found to modulate autophagy in human gastric and T-cell leukemia cells [18,31,32]. CucD has also been found to counteract chemoresistance; in particular, it suppressed the proliferation of gemcitabine-resistant pancreatic cancer cells [32]. Moreover, it was found to have a synergistic effect with cisplatin in human lung tumor cells that was characterized by inhibition of *p*-AKT, *p*-Erk, *p*-JNK, and *p*-ErbB3 signaling and suppression of STAT3 and NF-κB activity [33]. Similar effects were observed in doxorubicin-resistant human breast carcinoma cells [17]. Finally, CucD was found to overcome gefitinib resistance by modulating EGFR expression [16]. Surprisingly, most of the pathways affected by CucD are also related to the expression of the ZNF217 gene. Indeed, the ErbB3 gene is a direct target for ZNF217, and its overexpression determines the activation of both the PI3K/Akt and Ras/MAPK survival pathways [5,32,34]. Furthermore, EGF is implicated in ZNF217-induced immortality, and STAT3 seems to play a regulatory role in ZNF217 expression [34].

Based on the above findings, in the subsequent set of experiments, we investigated the effect of CucD on ZNF217 and found that CucD was able to significantly decrease ZNF217 expression in OCI-AML3 cells. This effect of CucD on ZNF217 expression was found to result from the inhibition of proliferation and not increase in apoptosis, as demonstrated by the results of gene silencing experiments in AML cells. Interestingly, this effect of CucD was not detected in other hematologic cancer cell lines. More importantly, CucD decreased ZNF217 expression in nucleophosmin (NPM)-mutated AML cells but not in AML cells with no NPM mutation. NPM mutations are present in about 30% of AML forms, so this finding opens the possibility that oncogenic expression of ZNF217 is linked to the NPM mutation. Despite notable advancements in the treatment of this frequent AML subtype in recent years, approximately 50% of patients with AML with the NPM1 mutation treated with conventional regimens eventually died as a result of disease progression. Given this poor survival outcome, it might be beneficial to explore strategies targeting ZNF217 to counteract AML with NPM1 mutations in order to increase patient survival. Thus, in the future, further investigations are required to confirm the present findings, to further understand the underlying mechanisms, and to evaluate the therapeutic possibilities of CucD in these forms of AML.

## 4. Materials and Methods

### 4.1. Plant Material and Isolation of Cucurbitacins

*E. hainanensis* Oliv. materials were collected from Ha Tinh province, Vietnam, in November 2021. The species was identified by Dr. Do Ngoc Dai, a botanist at the Faculty of Agriculture, Forestry and Fishery, Nghe An University of Economics. The voucher specimen (No. 04TN.EH21) was deposited at the Institute of Chemistry, VAST. The extraction and isolation of cucurbitacins are described in detail in the Appendix A Section.

### 4.2. Cell Line Culture and Characterization

AML (OCI-AML3), monocytic lymphoma (U937), acute T-cell lymphoblastic leukemia (Jurkat), and chronic B-cell lymphocytic leukemia (MEC1, PGA1) cell lines of human origin were obtained from ATCC (Manassas, VA, USA). OCI-AML3, U-937, HL-60, and MEC1 cells were maintained in RPMI-16140 medium, while PGA1 cells were maintained in IMDM medium. In both media, 10% fetal bovine serum, 100 U/mL penicillin, and 100 μg/mL streptomycin were used as cofactors. Cells were grown under standard conditions (37 °C and 5% CO_2_), and the media were changed twice per week, as recommended by the suppliers. When the cells reached approximately 80% confluency, they were sub-cultured. In the experiments, cells were seeded into 24-well plates, maintained at a concentration of 2 × 10^5^ cells/mL, and treated with different concentrations of dimethyl sulfoxide (DMSO) or the test compounds (0.03–10 μg/mL) for 24 h. DMSO was used at a maximum nontoxic concentration of 1% v/v in the medium.

### 4.3. Analysis of Cell Viability and Cell Cycle Progression

Cells were counted manually using a hemocytometer. As the cell count exceeded 200 in four chambers, it can be assumed that the margin of error was negligible. Cell viability and cell cycle progression were analyzed via flow cytometry to determine the DNA content of cell nuclei stained with PI after the exclusion of necrotic cells by forward light scatter. Cells were collected by centrifugation and washed in phosphate-buffered saline; DNA was stained by incubating the cells in H_2_O containing 50 μg/mL PI for 30 min at 4 °C. This allowed for direct DNA staining in PI hypotonic solution without the need for RNase treatment, as the RNA is eliminated by hypotonic shock. Fluorescence was measured by flow cytometry using Coulter Epics XL-MCL (Beckman Coulter Inc., Brea, CA, USA) and analyzed by the FlowJo_V10 software.

### 4.4. Western Blotting

Proteins were extracted with a RIPA buffer, separated by SDS-PAGE, and analyzed using Western blotting. The primary antibodies included anti-p21, anti-phospho-p38, anti-p38, anti-phospho-ERK, anti-ERK (Cell Signaling, Danvers, MA, USA), anti-p53 (Santa Cruz Biotechnology, Dallas, TX, USA), and anti-ZNF217 (Abcam, Cambridge, UK) antibodies. Anti-laminin (Sigma-Aldrich, St. Louis, MO, USA) and anti-GAPDH (OriGene, Rockville, MD, USA) antibodies were used as controls. Secondary antibodies were labeled with horseradish peroxidase (Pierce/Thermo-Fisher Scientific, Waltham, MA, USA). Antigen–antibody complexes were visualized by enhanced chemiluminescence according to the manufacturer’s instructions (Millipore, Billerica, MA, USA). Western blotting films were scanned and band signal intensities were determined using the ImageJ software https://imagej.net/ij/ (accessed on 25 August 2024) (National Institutes of Health, Bethesda, MD, USA).

### 4.5. Real-Time PCR

RNA was isolated using the Qiagen RNeasy Plus Micro kit, and conversion of total RNA to cDNA was performed with a QuantiTect Reverse Transcription kit (Qiagen, Hilden, Germany). Real-time PCR was performed with an ABI-7300 Real-Time Cycler (Applied Biosystems, Foster City, CA, USA), and amplification was achieved using a TaqMan Assay (Hs00998133m1 for *TGFb1*, Hs00919915_m1 for *Znf217*, hs00174128 m1 for *TNF-α*, and hs00187848_m1 for *Bcl2*; Thermo Fisher Scientific, Waltham, MA, USA). A ∆Ct method was used to determine the expression levels of *TNF-α*, *TGF-β*, *BCL-2*, and *ZNF217*.

### 4.6. ZNF217 Silencing

A Lipofectamine RNAiMax Reagent (Invitrogen, Carlsbad, CA, USA) was used for cell transfection with siRNA according to the manufacturer’s instructions. The two previously validated small interfering RNAs (siRNAs) targeting ZNF217 and a scrambled control (Life Technologies, Carlsbad, CA, USA) were transfected into the cells. OCI-AML3 cells were transfected with either ZNF217 or scrambled control siRNA in a culture medium for 24 h, 48 h, and 72 h. Serum-free Opti-MEM (Gibco, Carlsbad, CA, USA) was used to dilute target plasmids and Lipofectamine. The cells were then controlled for ZNF217 expression using Western blotting and stained with PI for flow cytometry analysis.

### 4.7. Statistical Analyses

All the data are expressed as mean ± standard error (SE) of at least two biological replicates, with at least two technical replicates per concentration performed. The statistical analyses were carried out using a GraphPad Prism™ software (Version 6.00; GraphPad Software Inc., San Diego, CA, USA). Statistical significance was determined using one-way ANOVA, as specified in the figure legends. Differences were considered statistically significant for the following *p* values: *p* < 0.05, *p* < 0.01, *p* < 0.001, and *p* < 0.0001.

## 5. Conclusions

In conclusion, the obtained results suggest that CucD is a promising molecule for the treatment of NPM-mutated AML, although further studies are needed to confirm these findings and investigate its mechanisms of action in more depth. In particular, in-depth studies on how cucurbitacins affect the ZNF217 pathway are required, starting with reverse docking studies to determine which molecules are bound by these substances. Additionally, we have not explored the synergistic effect of cucurbitacins with other drugs used in the treatment of leukemia, e.g., derivatives of podophylotoxin, so this could be a focus of future studies. Finally, the possibility of cucurbitacins exhibiting toxicity against normal cells must be considered and explored, since toxicity studies were not performed at this stage.

## Figures and Tables

**Figure 1 pharmaceuticals-17-01561-f001:**
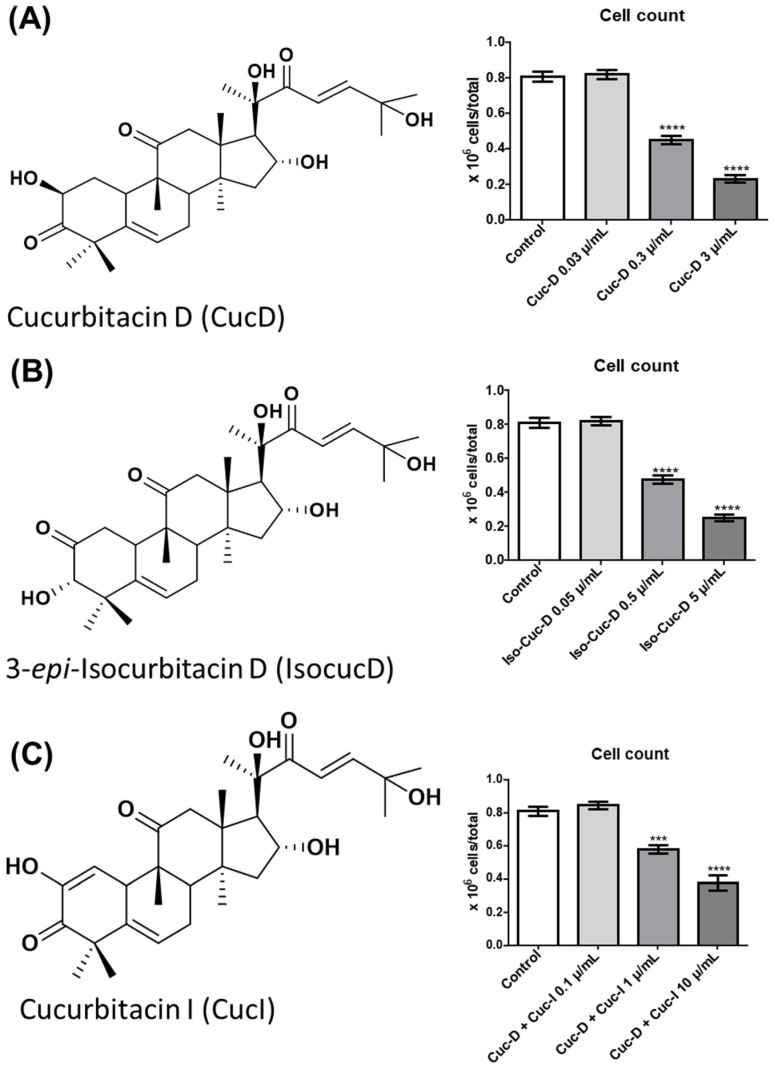
Effect of cucurbitacins on OCI-AML3 cell count. The panels on the left depict the chemical structures of cucurbitacin D (**A**), 3-epi-isocucurbitacin D (**B**), and cucurbitacin D + cucurbitacin I (**C**). In the panels on the right, the bars represent the number of viable cells counted after 24 h of treatment with control vehicle (Control), cucurbitacin D (**A**), 3-epi-isocucurbitacin D (**B**), or cucurbitacin D + cucurbitacin I (**C**) at the concentrations shown on the x-axis. The mean ± SEM values were determined from data from five independent experiments. *** *p* < 0.001 and **** *p* < 0.0001 are indicative of a significant decrease in cell viability in comparison to the control (calculated by one-way ANOVA).

**Figure 2 pharmaceuticals-17-01561-f002:**
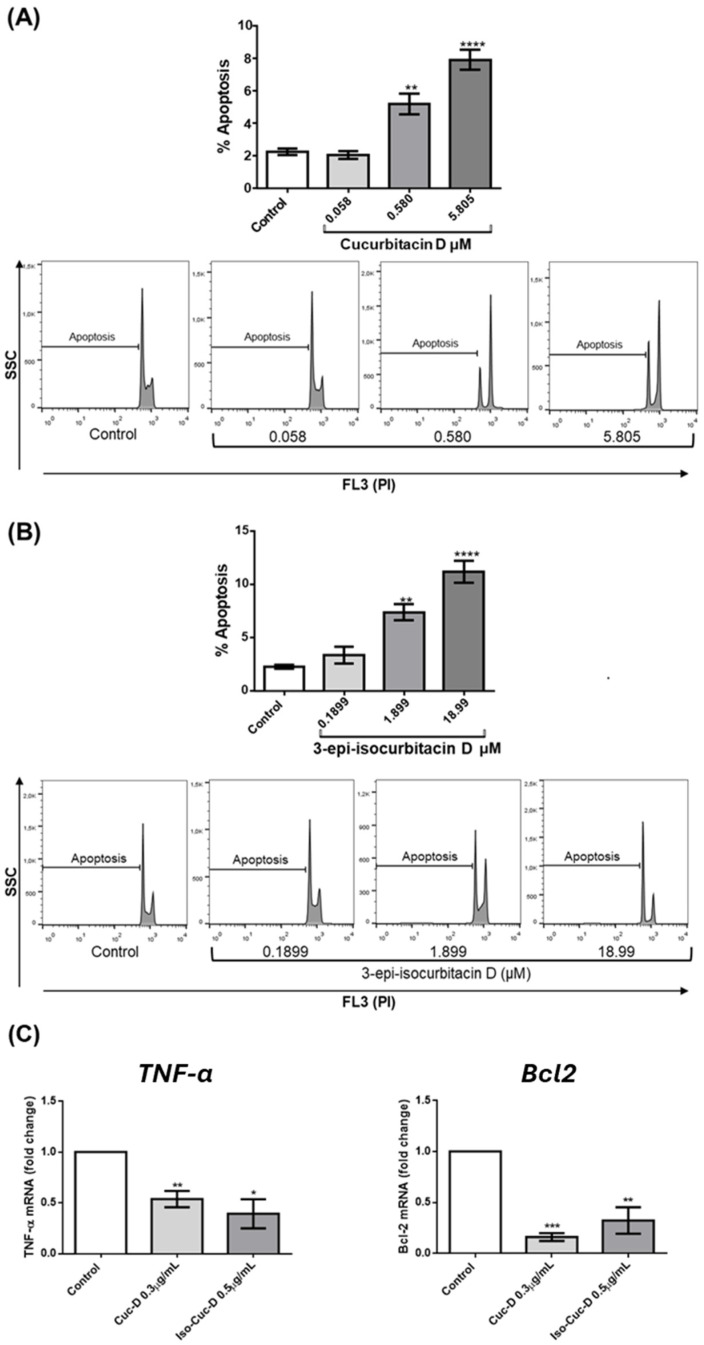
Effect of CucD and IsocucD on OCI-AML3 cell apoptosis. The bars represent the percentage of apoptotic cells after 24 h of treatment with the control vehicle (Control), CucD (**A**), or IsocucD (**B**), at the concentrations shown on the x-axis. The percentage of apoptotic cells was determined based on propidium iodide (PI) staining and is shown on the x-axis as the logarithmic scale values (FL3). (**C**) The bars represent the fold changes in TNF-α (left panel) and Bcl2 (right panel) after 24 h of treatment with the control vehicle (Control), CucD, or IsocucD at the concentrations reported on the x-axis. The graph shows the mean ± SEM values calculated from data from five independent experiments. * *p* < 0.05, ** *p* < 0.01, *** *p* < 0.001, and **** *p* < 0.0001 indicate a significant increase in the apoptosis rate in comparison to the control (calculated by one-way ANOVA).

**Figure 3 pharmaceuticals-17-01561-f003:**
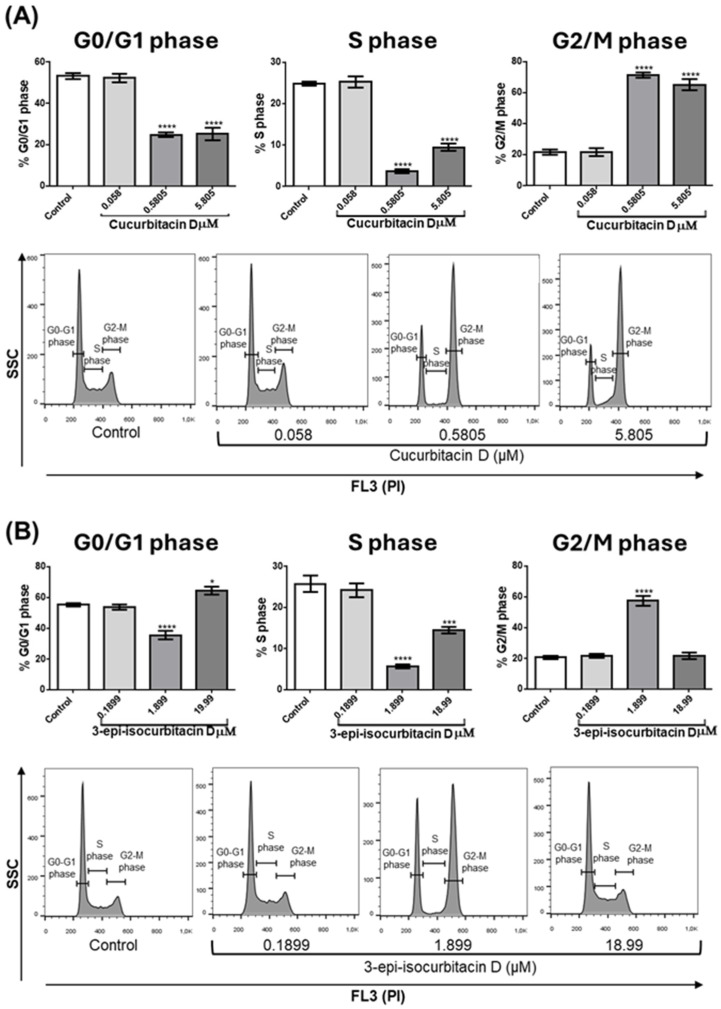
Effect of CucD and IsocucD on OCI-AML3 cell cycle progression. (**A**). The bars represent the percentage of cells in different phases of the cell cycle (left panels: G0/G1 phase, middle panels: S phase, right panels: G2/M phase) after 24 h of treatment with DMSO as the vehicle (Control), CucD (**A**), or IsoCucD (**B**) at the concentrations shown on the x-axis. Histograms from representative experiments show the values of PI staining on a logarithmic scale (FL2) on the X axes. The mean ± SEM values from five independent experiments are shown. * *p* < 0.05, *** *p* < 0.001, and **** *p* < 0.0001 indicate significant differences in comparison to the control group (calculated by one-way ANOVA).

**Figure 4 pharmaceuticals-17-01561-f004:**
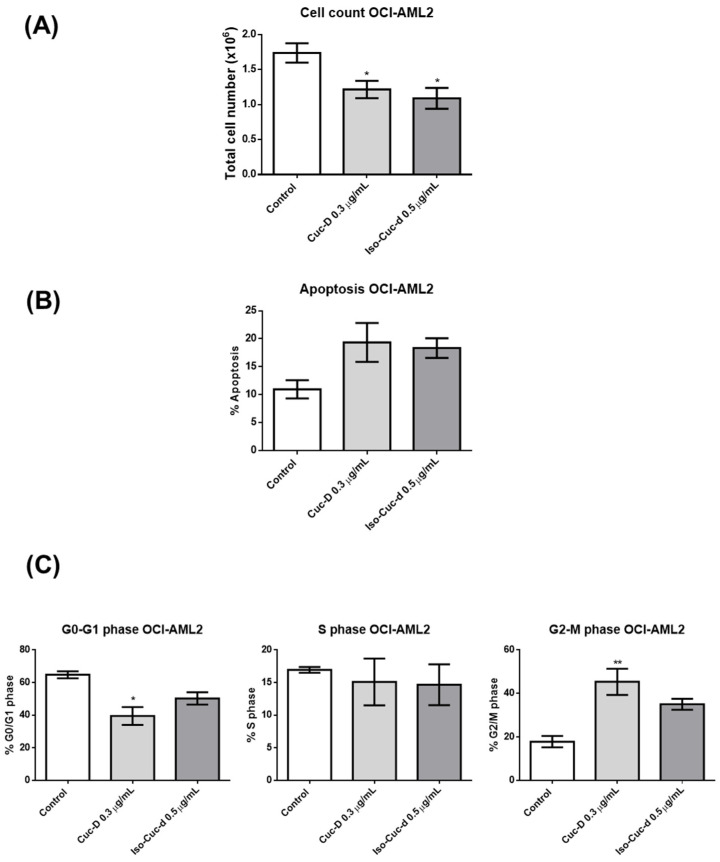
Effect of CucD and IsocucD on OCI-AML2. The bars represent the number of cells (**A**), the percentage of apoptotic cells (**B**), or, in (**C**), the percentage of cells in the G0/G1 (left), S (middle), or G2/M (right) phases of the cell cycle after 24 h of treatment with the control vehicle (Control), CucD, or IsoCucD, at the concentrations shown on the x-axis. The graphs show the mean ± SEM values calculated from data of three independent experiments. * *p* < 0.05 and ** *p* < 0.01 indicate a significant difference in comparison to the control (calculated by one-way ANOVA).

**Figure 5 pharmaceuticals-17-01561-f005:**
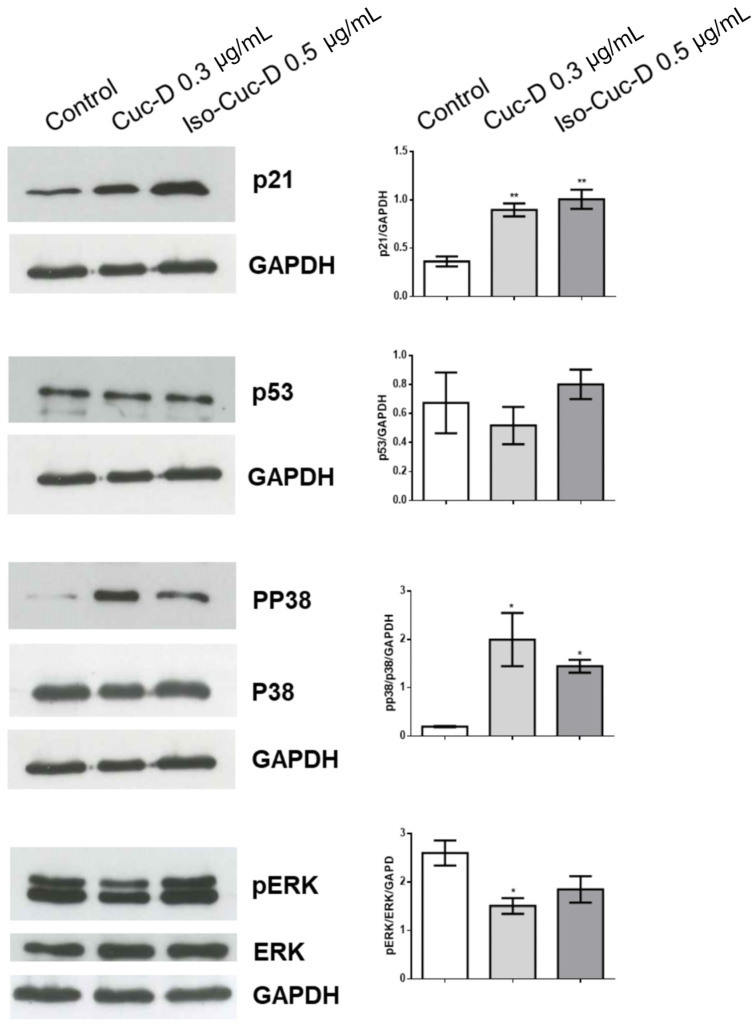
Effects of CucD and IsocucD on the expression of proteins involved in the cell cycle. Western blot illustrating the expression of p21 (in the first panel starting from the upper side), p53 (second line panels), phosphorylated p38 (pp38), total p38 (p38) (third line panels), phosphorylated ERK (pERK), and total ERK (ERK) (fourth line panels) in OCI-AML3 cells treated with vehicle (DMSO), CucD, or IsocucD for 24 h. The Western blots on the left side are representative of five independent experiments, and the corresponding data are quantified in the bar graphs on the right. GADPH served as a loading control. Data are reported as mean ± SEM. * *p* < 0.05, ** *p* < 0.01.

**Figure 6 pharmaceuticals-17-01561-f006:**
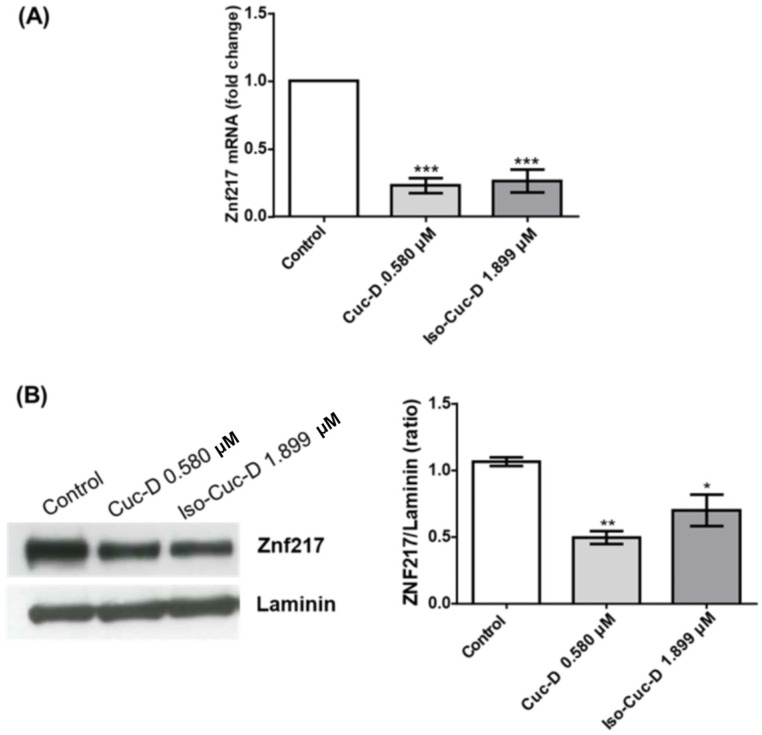
Effects of the tested compounds on ZNF217 expression in OCI-AML3 cells. (**A**) Real-time PCR of ZNF217 transcripts in OCI-AML3 cells treated with the vehicle (DMSO, Control), CucD, or IsocucD for 24 h. Gene expression was normalized to the expression of 18 S, and the normalized expression levels are reported (white bar, fold change = 1). The mean ± SEM values from five independent experiments are reported. *** *p* < 0.001 indicates significant differences in comparison to the control (calculated by one-way ANOVA). (**B**) Expression of the protein level of ZNF217 in OCI-AML3 cells treated with vehicle (DMSO, Control), Cuc-D, or IsocucD for 24 h. The expression level was normalized to that of laminin expression. The Western blot on the left is representative of four independent experiments, and the corresponding data are quantitatively analyzed in the right panel. The ZNF217/laminin ratio was calculated by densitometric quantification of the specific bands detected in four independent experiments. The mean ± SEM values from four independent experiments are reported. * *p* < 0.05, ** *p* < 0.01 and *** *p* < 0.01 indicate significant differences in comparison to the control group (calculated by one-way ANOVA).

**Figure 7 pharmaceuticals-17-01561-f007:**
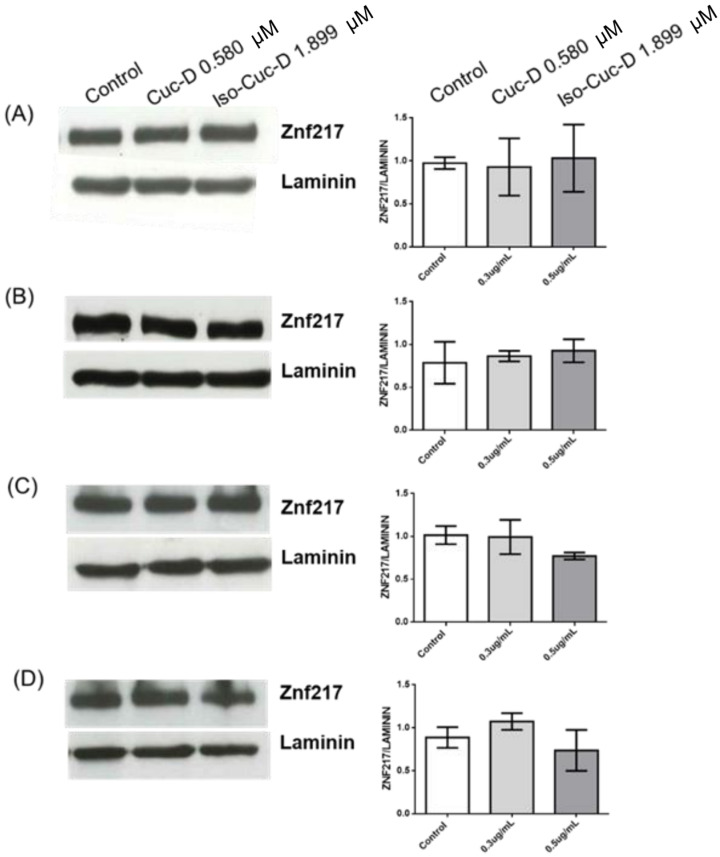
Expression of the protein level of ZNF217 in different cell lines treated with cucurbitacins. The PGA1 (**A**), MEC1 (**B**), U9370 (**C**), and Jurkat (**D**) cell lines were treated with vehicle (DMSO, Control), CucD, or IsocucD for 24 h. The expression level was normalized to that of laminin expression. The Western blots are representative of three independent experiments. The ZNF217/laminin ratio is calculated by densitometric quantification of the specific bands detected in three independent experiments. Data (mean ± SEM) are reported as fold change in ZNF217 protein expression in samples treated with vehicle (DMSO), CucD, or IsocucD.

**Figure 8 pharmaceuticals-17-01561-f008:**
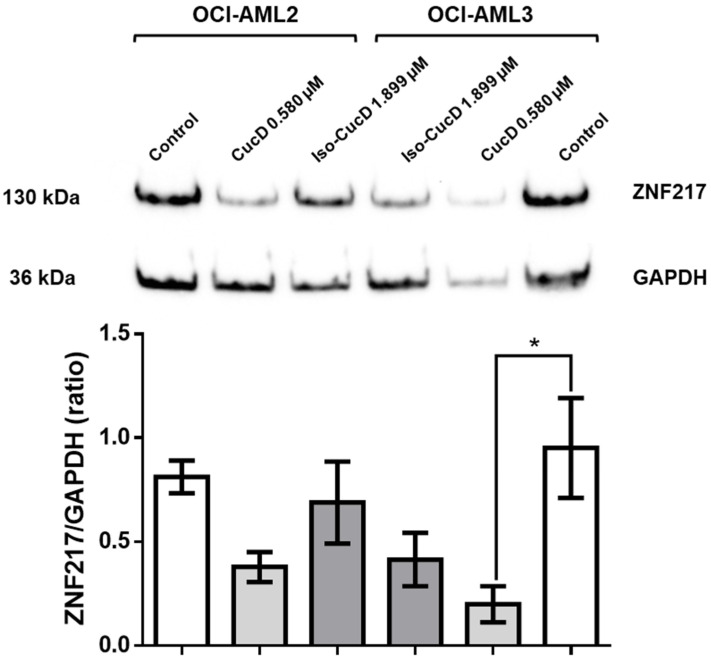
Comparison of the expression of the protein level of ZNF217 in OCI-AML2 and OCI-AML3 cell lines. OCI-AML2 and OCI-AML3 cell lines were treated with vehicle (DMSO, Control), CucD, or IsocucD for 24 h. Th expression level was normalized to that of GAPDH expression. The Western blots are representative of three independent experiments. The ZNF217/GAPDH ratio is calculated by densitometric quantification of the specific bands detected in three independent experiments. Data (mean ± SEM) are reported as fold change in ZNF217 protein expression in samples treated with vehicle (DMSO), CucD, or IsocucD. * *p* > 0.05.

**Figure 9 pharmaceuticals-17-01561-f009:**
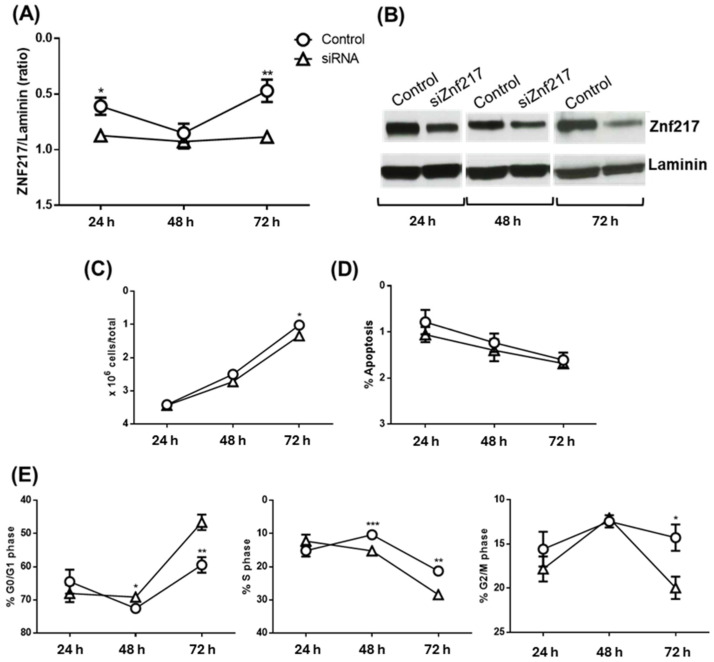
Effect of silencing ZNF217 on the viability of OCI-AML3 cells. (**A**) Expression of the ZNF217 protein in OCI-AML3 cells transfected with ZNF217 (siRNA) for 24 h, 48 h, and 72 h. ZNF217 expression was measured using Western blot analysis and normalized to laminin expression. (**B**) The Western blot is representative of five independent experiments. The ZNF217/laminin ratio was calculated by densitometric quantification of the specific bands detected in five independent experiments. Data (mean ± SEM) are reported as fold changes in expression in samples transfected with ZNF217 siRNA. * *p* < 0.05 and ** *p* < 0.01 indicate significant differences in comparison to the control (calculated by one-way ANOVA). (**C**) At 24 h, 48 h, and 72 h after treatment with the ZNF217 siRNA, the number of viable OCI-AML3 cells was determined by the trypan blue exclusion method. (**D**) Apoptosis and (**E**) cell cycle progression were evaluated by PI staining experiments. Data from five independent experiments are reported as mean ± SEM. * *p* < 0.05, ** *p* < 0.001, and *** *p* < 0.0001 indicate significant differences in comparison to the control group (calculated by one-way ANOVA).

## Data Availability

The original contributions presented in the study are included in the article/Appendix A, further inquiries can be directed to the corresponding author.

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
