# Peer review of "Modulatory Effect of Cucurbitacin D from Elaeocarpus hainanensis on ZNF217 Oncogene Expression in NPM-Mutated Acute Myeloid Leukemia"

_pharmaceuticals, 2024, doi:10.3390/ph17121561_

Round 1
Reviewer 1 Report
Comments and Suggestions for Authors The authors present the results on the role of Cucurbitacins on ZNF217 in OCI-AML cell line. It is an interesting study, however, there are several critical issues:- Authors used 1 cell line -OCI-AML3 for all experiments. It is hard to make any conclusions based on the results obtained from 1 cell line. Authors should repeat their experiments ( including apoptosis, cell cycle), using more AML cell lines to be able to make any conclusions on the role of Cucurbitacins in AML cell lines.
- In the title it says "NPM-mutated AML" when actually authors only worked with 1 AML cell line - OCI.
- ZNF217 expression was separately tested in 1 another AML cell line and 2 other hematologic cancers ( T cell leukemia and chronic lymphocytic leukemia) . Authors should test it on more AML cell lines before moving to other types of hematologic cancers.
- Lines 98-100: It is unclear how authors isolated specific Cucurbitacins and how they were characterised.
- The text requires extensive proofreading
Some sentences are unclear. Proof-reading is required
Author Response
- Comment #1: Authors used 1 cell line -OCI-AML3 for all experiments. It is hard to make any conclusions based on the results obtained from 1 cell line. Authors should repeat their experiments (including apoptosis, cell cycle), using more AML cell lines to be able to make any conclusions on the role of Cucurbitacins in AML cell lines.
- Comment #2: In the title it says “NPM-mutated AML” when actually authors only worked with 1 AML cell line – OCI.
- Comment #3: ZNF217 expression was separately tested in 1 another AML cell line and 2 other hematologic cancers (T cell leukemia and chronic lymphocytic leukemia). Authors should test it on more AML cell lines before moving to other types of hematologic cancers.
- Comment #4: Lines 98-100: It is unclear how authors isolated specific Cucurbitacins and how they were characterized.
- Comment #5: The text requires extensive proofreading.
Response:
- The reviewer raises a good point. At the beginning, we have utilized only the OCI-AML3 cell line because, to the best of our knowledge, this is the only knowm AML cell line that carry an NPM mutation and we have focused our work on the modulation of the ZNF217 expression. This is the reason why we have started with only OCI-AML3 and then we confirmed our results on ZNF217 expression also on additional 4 hematologic cancer cell lines. To meet the expectations of the reviewer, we have defrosted the acute myelois leukemia U937 cell line to perform experiments on apoptosis and cell cycle on an additional AML cell line. Furthermore we got the cell line OCI-AML2, which is a cell line similar to OCI-AML3 but without NPM mutation in order to perform additional experiments on apoptosis and cell cycle plus western blot for ZNF217 expression as asked by the reviewer. Unfortunatelly, the experiments are ongoing and the results are not ready within the time frame allowed for the revision.
- Response: As stated in point 1, OCI-AML3 is the only available AML cell line with NPM-mutation and this is the reason why, at the beginning, we have concentrated our efforts on this unique cell line.
- We have felt that working on 2 AML cell lines and on 3 additional hematologic cancers would have given a wider level of profundity to the study on expression of ZNF217. Again, making experiments on additional AML cell lines is not feasible within the revision time frame.
- The extraction and isolation of cucurbitacins have been described in detail in the Supporting Information. Cucurbitacins were characterized by NMR spectroscopic analysis and comparison with those of reported data, the structures of compounds 1- 3 were identified as cucurbitacin D, 3-epi-isocucurbitacin D and cucurbitacin I (see detailed NMR data in Table S1 in the Supporting Information). This is now further clarified in the text (lines 108-114).
- The text was sent for extensive proofreading by a professional agency.
Reviewer 2 Report
Comments and Suggestions for Authors
The article has a scientific merit and provides readers with new data related to the plant-based compounds and their potential use anticancer agents. It connects to modern trends in science in order to avoid synthetic, very often harmful compounds fused as anticancer drugs. In this case new approaches are needed and this article signs in that conception.
- 35. please do not use a capital letter in case of cucurbitacin D if it is put in the middle of sentence. Make a revision in the whole text.
- Elaeocarpus hainanensis - please add some information about use it in ethnopharmacology.
- did you try to stydu a synergism of these compounds with a drug used in the treatment of leukemia, eg. derivatives of podophylotoxin?
- 4.1.Pleae provide more information about plant material: You used a fresh plant or after drying? What type of extraction and solvent were applied?
Conclusion: - please re-phrase this part, try to avoid a repetition of results etc. in this part. Make a conclusion, what more may be done in the future, how the plant can be used in the industry. What about limitations?
Author Response
Comments:
-35. Please do not use a capital letter in case of cucurbitacin D if it is put in the middle of the sentence. Make a revision in the whole text.
- Elaeocarpus hainanensis – please add some information about use it in ethnopharmacology.
- did you try to study a synergism of these compounds with a drug used in the treatment of leukemia, eg. Derivatives of podophylotoxin?
- 4.1. Please provide more information about plant material: You used a fresh plant or after drying? Ehat type of extraction and solvent were applied?
Conclusion: - please re-phrase this part, try to avoid a repetition of results etc. in this part. Make a conclusion, what more may be done in the future, how the plant can be used in the industry. What about limitations?
Responses:
We thank the reviewer for the positive general comment.
-35. As suggested, we replaced the capital letter when cucurbitacin D it is put in the middle of the sentence. We have made a revision in the whole text.
- Unfortunatelly, there is a scarcity of information about the use of Elaecarpus hainanensis in ethnopharmacology and we have put in the text all information and cited articles where we found it. This is now specified in the text (lines 92-39).
- We haven’t tried to study the sinergism of cucurbitacins with a drug used in the treatment of leukemia, eg. Derivatives of podophylotoxin. This can be the objective of future studies and we now have added this possibility in the “conclusion” section (lines 470-472).
-4.1. More information about plant material has been added in “Results” and “materials and methods” section and exaustive information are present in “supplementary files” (lines 108-114; lines 396-398).
Conclusions. We have re-phrased this part according to suggestion of the reviewer (lines 467-474).
Reviewer 3 Report
Comments and Suggestions for Authors
In the manuscript by Adorisio et al., the authors aimed to evaluate the chemopreventive properties of cucurbitacin (Cuc) D and the modulation of ZNF217 in acute myeloid leukemia and some hematology cancer cell lines that express ZNF217. For this purpose, first, the authors isolated Cuc D, isocuc D, and Cuc I from the aerial parts of the Vietnamese plant Elaeocarpus hainanensis Oliv. Then, the authors evaluated the effects of purified compounds on viability, apoptosis, and cell cycle progression on nucleophosmin (NPM)-mutated acute myeloid leukemia (AML) and other hematology cancer cell lines. Finally, the effect of the different compounds on the expression of ZNF217 was also evaluated.
In general, the manuscript presents relevant data. The introduction clearly outlines the research objectives and current knowledge on the topic. Similarly, the materials and methods section is comprehensive but would benefit from more specific details and clarifications regarding some methods. However, the data treatment and presentation require a deep revision. Additionally, the authors should be cautious regarding some conclusions that are supported by the data. The discussion offers a good overview of the study’s findings and potential implications.
Based on my assessment, with further adjustments, the manuscript could be considered for publication.
Below are some suggestions to improve the manuscript.
General comment:
As I read through the manuscript, I noticed some minor grammatical errors and typos. These mistakes, although minor, can detract from the overall readability of the article. I recommend paying close attention to grammar, punctuation, sentence structure, and spelling to ensure a polished and error-free final version.
Also, consider defining the abbreviations when first cited (e.g., cucurbitacin).
Specific comments:
Abstract:
Line 21: “The expression of the oncogene zinc-finger protein 217 (ZNF217) has been reported to play a central role in cancer development, resistance, and recurrence.”
The authors should consider removing the word “central” as carcinogenesis is a multifactorial process, and a unique oncogene cannot be considered central.
Results section:
The authors should avoid some useless methodology description details in the results part. In the description result, consider go straight to the main point, this will enhance overall readability.
Line 115-119: “Since our compounds resulted in a positive effect, we only added negative control to avoid false positives. While we did not add positive controls, which were necessary if the results were negative (to avoid false negatives). Furthermore, being three, each compound acted as a positive control for the others.”
This is not necessary. That are the rudiments of experimental design and I feel then that these sentences should be removed.
The authors should avoid using the term “powerful”. Use “significant” instead.
Lines 128-130: Due to the low level of apoptosis, its increase can only be detected by measurement and is not detectable by eyes.
This sentence is not necessary. Please remove this sentence. Furthermore, even if you notice, a significant decrease in cell number you cannot claim if it is apoptosis, necrosis (or another type of cell death) or inhibition of cell proliferation. The unique way is to measure it.
Line 133-139: “Apoptosis is activated by at least two different pathways. The mitochondrial (i.e., intrinsic) pathway leads to down-regulation of anti-apoptotic molecules, such as Bcl2, with the following sequential release of cytochrome c from mitochondria and activation of caspase-9, which directly cleaves and activates caspase-3. The second (i.e., extrinsic) pathway involves activation of caspase-8 that is triggered by the stimulation of death receptors such as TNFR by its ligand TNF-α.”
I feel that this information is not necessary here in the results section.
Line 141-142, Lines 144-145: “Figure 2 shows that the pro-apoptotic molecule TNF-α was significantly down-regulated”. “These results indicate that the TNF-α-dependent extrinsic apoptotic pathway was inhibited by CucD and IsoCucD treatment.”
Since the authors evaluated the levels of TNF-α mRNA after 24h of treatment, they cannot claim that the pathway was inhibited. As the extrinsic pathway is triggered by TNF-α (or other death ligands), they should have evaluated its expression at early stages after the treatment. This down-regulation may be the result of the apoptosis process. Furthermore, the authors evaluated mRNA levels and not protein expression. mRNA level and protein expression don’t necessarily match. If the aim of the authors was to evaluate whether the extrinsic pathway was activated, they should have assessed caspase-8 activation.
Lines 171-174: “Because p21 is regulated by p53 [18, 19], we also investigated whether CucD or iso CucD treatment induces p53 expression in OCI-AML3 cells. We found that both compounds did not affect the expression of p53, suggesting that p21-dependent, p53-independent pathway is at least partially involved for the effect of CucD and IsoCucD concentrations on cell cycle arrest.”
The conclusion is not supported by the data presented. The authors evaluated p53 expression while they should have assessed its activation. The expression may not change while the state of activation does. The authors should be cautious in their conclusions.
Lines 179-180: “The presence of two bands in phosphorylated ERK is the result of different levels of phosphorylation”.
That are two isoforms of ERK (1/2) that are respectively 41/43 KDa when non-phosphorylated and 42/44 KDa when phosphorylated.
Materials and methods section:
The authors should include a section on the preparation of the extract from plant material and the subsequent isolation of the compounds.
The authors should add the sequences of the primers used for PCR.
The authors should provide the references of the antibodies used.
Conclusion
This section would benefit from additional context and suggestions for future research to enhance its completeness and impact.
Comments on the Quality of English LanguageModerate english editing is needed particularly for those typos and grammatical errors.
Author Response
Comment #1: As I read through the manuscript, I noticed some minor grammatical errors and typos. These mistakes, although minor, can detract from the overall readibility of the article. I recommend paying close attention to grammar, punctuation, sentence structure, and spelling to ensure a polished and error-free final version.
Also, consider defining the abbreviations when first cited (e.g., cucurbitacin).
Response: thanks for the remarks. The text has been now revised by a professional editing agency in order to avoid errors.
As suggested, abbreviation has been defined when first cited
Specific comments:
Abstract:
Line 21. “The expression of the oncogene zinc-finger protein 217 (ZNF217) has been reported to play a central role in cancer development, resistance, and recurrence.” The authors should consider removing the word “central” as carcinogenesis is a multifactorial process, and a unique oncogene cannot be considered central.
Response: As suggested, the word “central” has been removed.
Results section:
The authors should avoid some useless methodology description details in the results part. In the description result, consider go straight to the main point, this will enhance overall readibility.
Response: As suggested, useless methodology description details in the results part were eliminated.
Lines 115-119: “Since our compounds resulted in a positive effect, we only added negative controls to avoid false positives. While we did not add positive controls, which were necessary if the results were negative (to avoid false negatives). Furthermore, being three, each compound acted as a positive control for the others.”
This is not necessary. That are rudiments of experimental design and I feel then that these sentences should be removed.
Response: As suggested, the sentences has been removed.
The authors should avoid using the term “powerful”. Use “significant” instead,
Response: As suggested the word “powerful” has been replace with “significant”.
Lines 128-130: Due to the low level of apoptosis, its increase can only be detected by measurement and it is not detectable by eyes.
This sentence is not necessary. Please remove this sentence. Furthermore, even if you notice, a significant decrease in cell number you cannot claim if it is apoptosis, necrosis (or another type of cell death) or inhibition of cell proliferation. The unique way is to measure it.
Response: As suggested the sentence has been removed. We have measured apoptosis by PI and cytofluorimetric analysis (Analysis of apoptosis by propidium iodide staining and flow cytometry, Carlo Riccardi & Ildo Nicoletti, Nature Protocols volume 1, pages1458–1461 (2006)) that now we have added to reference list.
Lines 133-139: “Apoptosis is activated by at least two different pathways. The mitochondrial (i.e. intrinsic) pathway leads to down-regulation of anti-apoptotic molecules, such as Bcl2, with the following sequential release of cytochrome c from mitochondria and activation of caspase-9, which directly cleaves and activates caspase-3. The second (i.e. extrinsic) pathway involves activation of caspase-8 that is triggered by the stimulation of death receptors such as TNFR by its ligand TNF-α.”
I feel that this information is not necessary here in the results section.
Response: As suggested, theìis information has been moved in the “discussion” section.
Lines 141-142. Lines 144-145: “Figure 2 shows that the proapoptotic molecule TNF-αwas significantly down-regulated”. “These results indicate that the TNF-α-dependent extrinsic apoptotic pathway was inhibited by CucD and IsoCucD treatment.”
Since the authors evaluated the levels of TNF-αmRNA after 24 h of treatment, they cannot claim that the pathway was inhibited. As the extrinsic pathway is triggered by TNF-α (or other death ligands), they should have evaluated its expression at early stages of treatment. This down-regulation may be the result of the apoptosis process. Furthermore, the authors evaluated mRNA levels and not protein expression. mRNA level and protein expression don’t necessary match. If the aim of the authors was to evaluate whether the extrinsic pathway was activated, they should have assessed caspase-8 activation.
Response: I agree with the reasoning of the reviewer. This is now well explained in the discussion (lines 328-332, and 336-339)
Lines 171-174: “Because p21is regulated by p53 [18, 19], we also investigated whether CucD or iso CucD treatment induces p53 expression in OCI-AML3 cells. We found that both compounds did not affect the expression of p53, suggesting that p21-dependent, p-53-independent pathway is at least partially involved for the effect of CucD and IsoCucD concentrations on cell cycle arrest.”
The conclusions not supported by the data presented. The authors evaluated p53 expression while they should have assessed its activation. The expression may not change while the state of activation does. The authors should be catious in their conclusions.
Response: As suggested, the “p53-independent” word has been eliminated in order to meet the suggestion to be more cautious in our conclusions.
Lines 179-180: “The presence of two bands in phosphorylated ERK is the result of different levels of phosphorylation”.
That are two isoforms of ERK (1/2) that are respectively 41/43 KDa when non-phosphorylated and 42/44 KDa when phosphorylated.
Response: this information was now added to the text (lines 176-177).
Materials and methods section:
The authors should include a section on the preparation of the extract from plant material and the subsequent isolation of the compounds.
The authors should add the sequences of the primers used for PCR.
The authors should provide the references of the antibodies used.
Responses: The preparation of the extract from plant material and the subsequent isolation of the compounds is quite complex and it is described in details in supplementary files. This is now stated in the M&M section.
The sequences of TaqMan probes are proprietary to the manufacturer and are not disclosed to the public. They are not provided on the purchase website, as they are standard commercial products and not designed based on specific requestswe have modified the section according to the reviewer’s suggestions. In M&M section the commercial identification number of each primer is provided.
As suggested the references of the antibodies used were added except for ZNF217 antibody (the reference was not present in the product sheet).
Conclusion
This section should benefit from additional context and suggestions for future research to enhance its completeness and impact.
Response: The section has been modified as suggested.
Moderate english editing is needed particularly for those typos and grammatical errors.
Response: As suggested, the manuscript has been sent for revision to a professional editing agengy.
Round 2
Reviewer 1 Report
Comments and Suggestions for Authors
Authors have not addressed the comments due to the short revision time. I suggest to perform experiments with additional cells lines, which are critical for the publication of results.
Author Response
Comment 1: Authors have not addressed the comments due to the short revision time. I suggest to perform experiments with additional cells lines, which are critical for the publication of results.
Response: the editor added additional 5 days for the second round of revision. One more experiment (we were not able to perform more than one experiment within the 5 days time for round 2 revision) was performed on the additional acute myeloid leukemia cell line OCI-AML2 and on the monocytic lymphoma U937. The results of this experiment confirmed the decrease in cell number (we have performed 2 experiments on cell counts with OCI-AML2 cells), the increase of apoptosis and the block of cell cycle progression seen with the OCI-AML3 cells. This is now reported in the text of the revised paper (lines 132-135; 155-156; 179-181). Since we did not perform a number of independent experiments useful to calculate the significance due to lack of time for revision, the results were added as tables in "supplentary files".
Reviewer 3 Report
Comments and Suggestions for Authors
I went through the revised of the manuscript and I noticed that the authors addressed all the issues raised. The manuscripthas been greatly improved and it's now acceptable for publication in its current form.
Author Response
We thank the reviewer.
Round 3
Reviewer 1 Report
Comments and Suggestions for Authors
The authors did not have sufficient number of replicates to draw any conclusion. Sufficient replicates are required to publish results. The results should also be combined rather than presented in the Supplementary material.
Author Response
1) 3 independent experiments of cell counts, apoptosis and cell cycle analysis on a new acute leukemia cell line (OCI-AML2). We added a new figure 4 and a new paragraph in the text describing the reported experiments;
2) we performed 3 additional independent experiments of western blot comparing the expression of ZNF217 on both OCI-AML3 and OCI-AML2 under a 24 hours stimulation with CucD or IsoCucD. Notably, both substances were unable to significantly down-regulate ZNF217 in NPM-wild type OCI-AML2, showing a selectivity towards the NPM-mutated AML cell line. These experiments were added as a new figure 8 in the revised paper.
Round 4
Reviewer 1 Report
Comments and Suggestions for Authors
The comments were addressed by the authors